EMBO
Molecular Medicine

# Small molecule inhibitors of the Dishevelled-CXXC5 interaction are new drug candidates for bone anabolic osteoporosis therapy

Hyun-Yi Kim[1,2], Sehee Choi[1,2], Ji-Hye Yoon[3], Hwan Jung Lim[4], Hyuk Lee[4], Jiwon Choi[5], Eun Ji Ro[1,2], Jung-Nyoung Heo[4], Weontae Lee[3], Kyoung Tai No[2,5] & Kang-Yell Choi[1,2,*]

## Abstract

Bone anabolic agents promoting bone formation and rebuilding damaged bones would ideally overcome the limitations of anti-resorptive therapy, the current standard prescription for osteoporosis. However, the currently prescribed parathyroid hormone (PTH)-based anabolic drugs present limitations and adverse effects including osteosarcoma during long-term use. Also, the antibody-based anabolic drugs that are currently being developed present the potential limits in clinical application typical of macromolecule drugs. We previously identified that CXXC5 is a negative feedback regulator of the Wnt/β-catenin pathway via its interaction with Dishevelled (Dvl) and suggested the Dvl–CXXC5 interaction as a potential target for anabolic therapy of osteoporosis. Here, we screened small-molecule inhibitors of the Dvl–CXXC5 interaction via a newly established *in vitro* assay system. The screened compounds were found to activate the Wnt/β-catenin pathway and enhance osteoblast differentiation in primary osteoblasts. The bone anabolic effects of the compounds were shown using *ex vivo*-cultured calvaria. Nuclear magnetic resonance (NMR) titration analysis confirmed interaction between Dvl PDZ domain and KY-02061, a representative of the screened compounds. Oral administration of KY-02327, one of 55 newly synthesized KY-02061 analogs, successfully rescued bone loss in the ovariectomized (OVX) mouse model. In conclusion, small-molecule inhibitors of the Dvl–CXXC5 interaction that block negative feedback regulation of Wnt/β-catenin signaling are potential candidates for the development of bone anabolic anti-osteoporosis drugs.

**Keywords** CXXC5; Dishevelled; negative feedback regulation; osteoporosis; Wnt/β-catenin pathway
**Subject Categories** Musculoskeletal System; Pharmacology & Drug Discovery

## Introduction

Bone is seamlessly being created and replaced via a dynamic process, called bone remodeling. Bone remodeling consists of two sequential phases, bone resorption by osteoclasts and new bone formation by osteoblasts (Martin, 2014). These two phases must be in balance to maintain homeostasis and a constant bone mass. Osteoporotic (low bone mass) and osteopetrotic (high bone mass) diseases are caused by collapse of the balance (Lazner *et al*, 1999). While osteopetrotic diseases are rare, the prevalence of osteoporosis in postmenopausal women and aged men makes osteoporotic disease one of the major medical problems of modern society (Lazner *et al*, 1999).

The historical approach for osteoporotic disease treatment has been mainly focused on inhibition of bone resorption. However, the anti-resorptive osteoporosis therapies maintain bone mineral density (BMD) by inhibiting bone resorption, not by promoting new bone formation, suppressing elimination of the damaged old bone (Martin, 2014). Therefore, the long-term use of the anti-resorptive agents increases the risk of atypical fractures due to accumulation of micro-damage in bone (Geissler *et al*, 2015).

Bone anabolic therapy, a relatively recent approach, aims to promote bone formation and rebuild damaged bones, thus holding promise for the treatment of osteoporosis (Ishtiaq *et al*, 2015). Currently, full-length and N-terminal (1–34) fragments of the parathyroid hormone (PTH) are the only approved bone anabolic agents in clinical use (Ishtiaq *et al*, 2015). PTH stimulates survival signals of osteoblasts followed by the increase of osteoblast numbers and bone formation rate (Jilka, 2007). However, PTH-based drugs are relatively expensive compared with small-molecule drugs and require daily subcutaneous administration (Rachner *et al*, 2011). Hypercalcemia, an adverse effect of PTH treatments, and concerns about osteosarcoma development are also limitations for PTH therapies (Canalis *et al*, 2007).

1  Translational Research Center for Protein Function Control, Yonsei University, Seoul, Korea
2  Department of Biotechnology, College of Life Science and Biotechnology, Yonsei University, Seoul, Korea
3  Department of Biochemistry, College of Life Science and Biotechnology, Yonsei University, Seoul, Korea
4  Korea Research Institute of Chemical Technology, Daejeon, Korea
5  Bioinformatics & Molecular Design Research Center, Yonsei University, Seoul, Korea
   *Corresponding author. Tel: +82 22123 2887; Fax: +82 2123 8284; E-mail: kychoi@yonsei.ac.kr

Over the last few decades, remarkable progress has been achieved in the molecular study of bone and major signaling pathways regulating bone anabolism have been elucidated. Among them, the Wnt/β-catenin pathway is receiving attention as a potential target for the development of anabolic agents (Rachner et al, 2011; Long, 2012; Regard et al, 2012). This pathway is activated by the binding of secreted Wnt ligands to Frizzled (Fz) receptors, as well as to lipoprotein receptor-related protein 5 and 6 (LRP 5/6) co-receptors. Dishevelled (Dvl), a cytosolic component of the Wnt/β-catenin pathway, transmits the signal from the Wnt/Fz interaction to β-catenin by disrupting β-catenin interaction with a complex including APC, axin, and GSK3β. Once β-catenin is released from this complex, it translocates into the nucleus to activate downstream target genes (Regard et al, 2012). The nuclear accumulation of β-catenin is a major trigger of osteoblast differentiation and bone formation (Long, 2012).

Other components of this pathway are also known to regulate development and homeostasis of bone (Regard et al, 2012). Dickkopf-1 (Dkk-1) and sclerostin, two extracellular negative regulators of the Wnt/β-catenin pathway, are considered as potential targets for bone anabolic anti-osteoporosis therapy (Diarra et al, 2007; Fulciniti et al, 2009; Heath et al, 2009; Li et al, 2009; Ominsky et al, 2010). Antibodies targeting these two proteins showed notable bone anabolic effects in rat and primate osteoporotic models (Fulciniti et al, 2009; Ominsky et al, 2010), and sclerostin antibodies are currently under clinical trials as anti-osteoporosis agents (Ferrari, 2014). However, macromolecule drugs such as hormone- or antibody-based drugs are expensive and unavailable for oral administration. Therefore, small-molecule drugs targeting this pathway would be desirable for the development of anabolic anti-osteoporosis drugs.

In a previous study, we suggested the Dvl–CXXC5 interaction as a negative feedback mechanism of the Wnt/β-catenin pathway and as a potent target for bone anabolic agents (Kim et al, 2015). We confirmed this hypothesis using a newly synthesized competitor peptide of the Dvl–CXXC5 interaction, which induced osteoblast differentiation and enhanced bone formation of ex vivo-cultured calvariae (Kim et al, 2015). Here, we identified KY-02061, a small-molecule inhibitor of Dvl–CXXC5 interaction by establishing a new Dvl–CXXC5 in vitro binding assay system and verified the competitor peptide-mimicking effects of this compound. Similar binding patterns of the competitor peptide and KY-02061 to the Dvl PDZ domain were revealed by structural analyses using nuclear magnetic resonance spectroscopy (NMR). We synthesized 55 KY-02061 derivatives and selected KY-02327, a compound optimized for both stability and activity. This compound rescued decreases of BMD and trabecular number in ovariectomized (OVX) mice, the animal model for postmenopausal osteoporosis (Thompson et al, 1995), when orally administered. We conclude that small-molecule inhibitors of the Dvl–CXXC5 interaction are potential candidates for new bone anabolic anti-osteoporotic drugs.

# Results

## Small-molecule compounds blocking Dvl–CXXC5 interaction activated the Wnt/β-catenin signaling and enhanced ex vivo-cultured calvaria growth

We tried to search small-molecule drug candidates activating the Wnt/β-catenin signaling on the basis of the previous study (Kim

et al, 2015). To screen small molecules inhibiting the Dvl–CXXC5 interaction, an in vitro assay system (Fig 1A) was established using purified recombinant Dvl PDZ domain and FITC-conjugated PolyR-DBM (Dvl binding motif) (Kim et al, 2015). A chemical with inhibitory activity of Dvl–CXXC5 interaction competes with the FITC-conjugated peptide for binding to Dvl PDZ domain, which results in a decrease in fluorescence intensity of the well containing the chemical (Fig 1A).

Thousands of compounds from a commercial library and an in-house collection of Korea Chemical Bank in Korea Research Institute of Chemical Technology were screened using the assay system (Appendix Fig S1A). We found that several top-ranked lead compounds with inhibitory activity contain an indole ring in common. The small molecules with an indole ring or a carboxylic acid group have been reported to bind to the Dvl PDZ domain (Mahindroo et al, 2008; You et al, 2008; Lee et al, 2009). We examined several indole-containing chemicals selected from the in-house collection for further investigations (Fig 1B). Six out of 20 compounds in this group reduced the fluorescence intensity. KY-02052 and KY-02061, the most active compounds, reduced the fluorescence intensities by 45.1% and 43.7%, respectively (Fig 1B). No compounds in this group showed significant auto-fluorescence, which can produce distorted results (Appendix Fig S1B).

We previously observed the activation of Wnt/β-catenin pathway in primary osteoblast cells, in which the Dvl–CXXC5 interaction was inhibited using the competitor peptide (Kim et al, 2015). Similarly, treatment of the 6 candidate compounds increased nuclear accumulation of β-catenin, a hallmark of activated Wnt/β-catenin signaling (Fig 1C and D). KY-02061 was especially effective as revealed by the strong accumulation of β-catenin in cell nuclei, without any cytotoxic effect (Fig 1C and Appendix Fig S2A). Although KY-02052 significantly increased nuclear β-catenin, it showed severe cytotoxic effects as indicated by the reduction of cell number and viability (Fig 1C and Appendix Fig S2A). Alkaline phosphatase (ALP) activity, a reliable indicator of osteoblast differentiation (Wang et al, 1999), was increased by treatment with KY-02052, KY-02057, or KY-02061 in the primary osteoblast cells (Appendix Fig S2B). KY-02061 treatment led to the strongest increase of ALP activity, whereas KY-02052 showed severe toxic effects as indicated by increased number of floating cell debris (Appendix Fig S2B).

The calvaria extracted from postnatal day 4 mice grows in thickness during ex vivo culture (Reynolds et al, 1972). KY-02061 most significantly increased the thickness of ex vivo-cultured calvaria by 51.2% compared with the DMSO-treated control group (Fig 1E and F). Overall, KY-02061 showed the best properties among the tested small molecules, and we selected this compound for further investigation.

## KY-02061 interacted with the Dvl PDZ domain similarly with DBM and enhanced osteoblast differentiation

From commercially available 5-benzyloxy indole-2-carboxylate ethylester, KY-02061 was formally synthesized (Appendix Scheme S1). KY-02061 is composed of an indole ring with one 4-methylbenzene sulfonate group at position 5, a carboxylic acid ethylester group at position 2, and ethylacetate group at position 1 (Fig 2A). KY-02061 decreased Dvl–CXXC5 interaction in a dose-dependent manner as shown by the in vitro binding assay (Fig 1A) with 50% inhibition

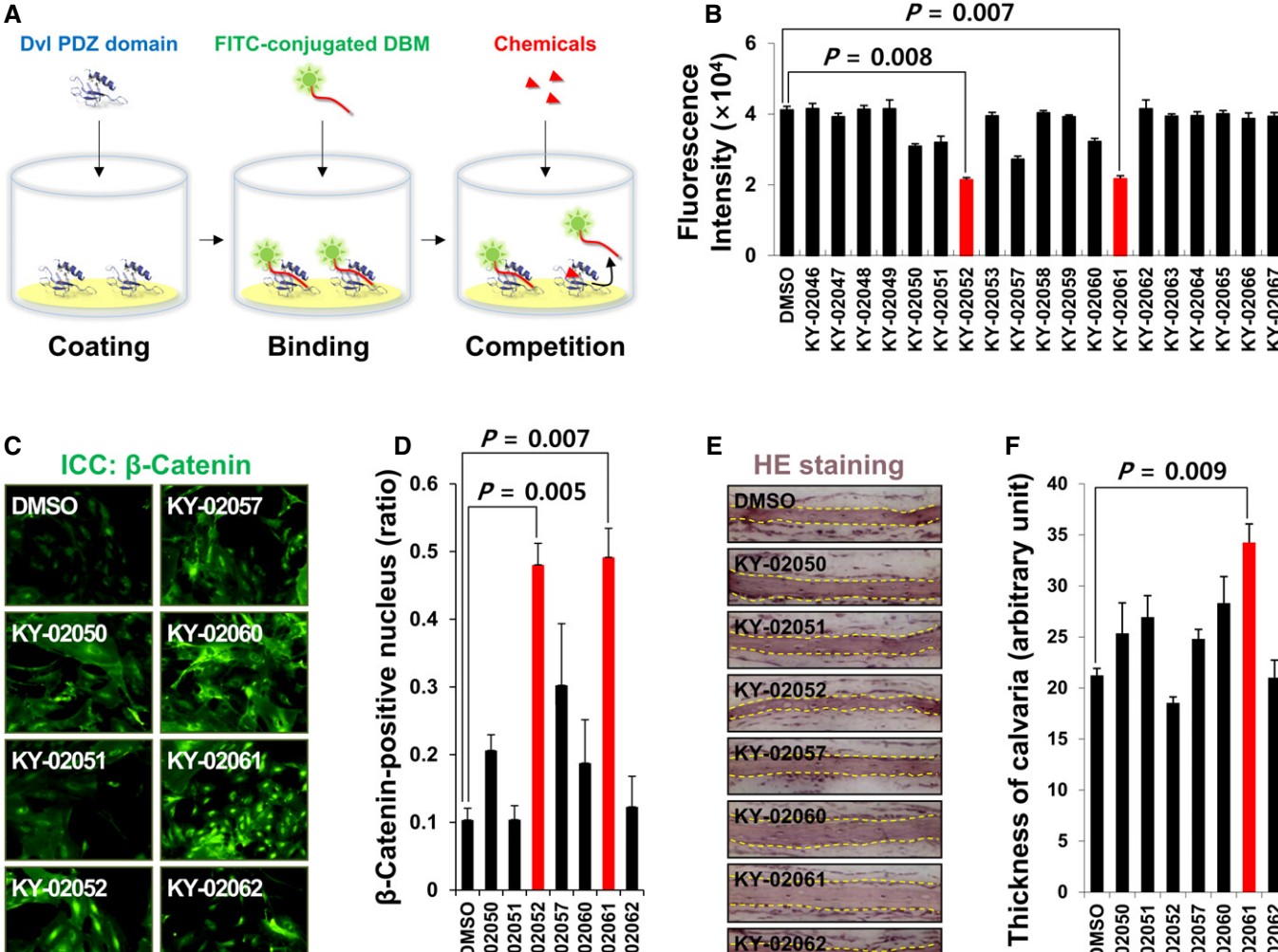

**Figure 1.  Identification and characterization of small molecules competing Dvl–CXXC5 binding in vitro.**

A    A scheme for *in vitro* screening method of small molecules competing Dvl–CXXC5 binding. Briefly, purified Dvl PDZ domain was attached to the polystyrene surface of each well of 96-well plates. Then, PolyR-DBM (polyarginine-conjugated Dvl binding motif tagged with FITC) (Kim *et al*, 2015) was added to each well and allowed to bind to the Dvl PDZ domain; 10 μM small-molecule compound was added to each well, and the compounds competing with Dvl PZD-PolyR-DBM binding were measured by a microplate reader to screen the compounds reducing fluorescence signal.

B    Screening results of compounds competing Dvl–CXXC5 interaction. Each 2 μM of the compounds was used for competing as described in Fig 1A. (*n* = 3)

C, D    Primary calvaria cells were treated with 2 μM of each compound for 2 days. The cells were subjected to immunofluorescence analyses to visualize β-catenin (C; green). The relative numbers of β-catenin-positive nuclei were counted (D). (*n* = 3)

E, F    The calvariae from 4-day-old mice were cultured *ex vivo* for 7 days with 2 μM of each compound. Representative calvaria sections were visualized by H&E staining (E). The calvaria thicknesses were measured from the images using Image Pro software (F). (*n* = 3)

Data Information: For B, D, and F, the data are the mean ± s.d. (error bars), and significance was assessed using unpaired Student's *t*-test.

concentration ($IC_{50}$) value of 24 μM (Fig 2B and Appendix Table S1).

To compare the binding patterns of the competitor peptide DBM and KY-02061 to the Dvl PDZ domain, titration experiments were performed using NMR spectroscopy. DBM and KY-02061 binding both induced chemical shifts of several residues on the Dvl PDZ domain (Appendix Fig S3A–C and Fig 2C–E). The Dvl PDZ domain comprises 6 β-sheet (βA-βE) and 2 α-helix (αA and αB) (Lee & Zheng, 2010). In the Dvl PDZ domain, three residues in βB (S265, I266, and V267), one residue in βC (I278), and three residues in αB

(L321, R322, and V325) were perturbed in the DBM interaction (Appendix Fig S3A–C). *In silico* molecular docking analysis showed that the carboxyl terminus of DBM fitted into a groove flanked by αB and β-sheet complex and interacted with the residues on the domains (Appendix Fig S3D and E).

In the KY-02061 interaction, one residue in βB (S265) and three residues in αB (L321, R322, and V325) were perturbed (Fig 2C–E). Four of the residues perturbed in the Dvl PDZ domain-DBM interaction (S265, L321, R322, and V325) were also perturbed in the KY-02061 interaction, which shows that KY-02061 binds to Dvl in

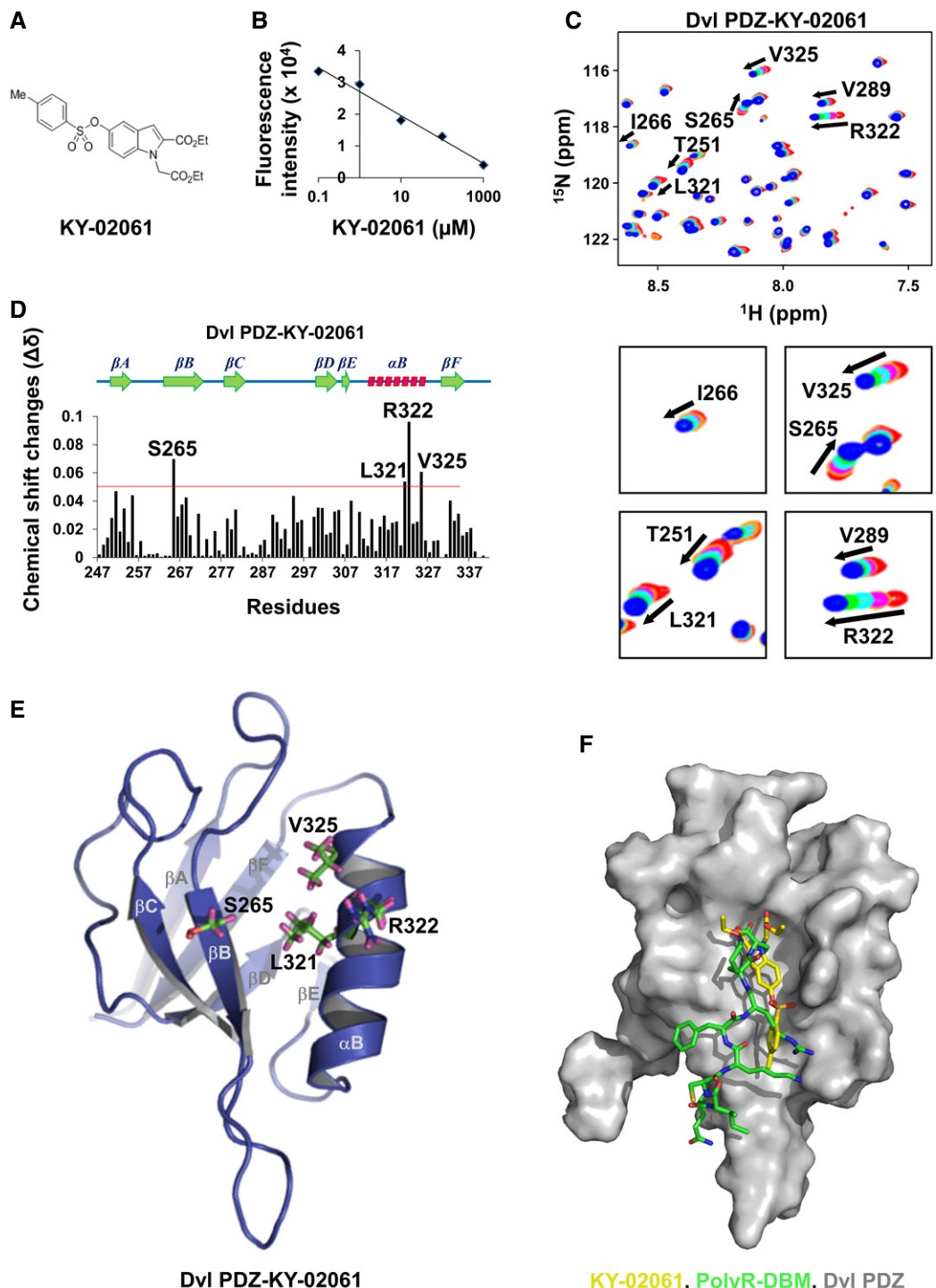

**Figure 2. DBM-mimetic binding of KY-02061 on the Dvl PDZ domain.**

A    The chemical structure of KY-02061.

B    A competition curve for the Dvl–CXXC5 interaction by KY-02061.

C–E    NMR titration analyses for Dvl PZD domain with KY-02061. $^1$H-$^{15}$N-HSQC analyses were performed to analyze the interaction of $^{15}$N-labeled Dvl PDZ domain with KY-02061. The $^1$H-$^{15}$N-HSQC spectrum of different molar ratios (Dvl PDZ domain:KY-02061) is displayed as red (1:0), orange (1:10), purple (1:20), cyan (1:40), green (1:60), and blue (1:80) (C, residues with meaningful chemical shift change are indicated by arrows). Plot of chemical shift changes (Δδ) as a function of residue number in molecular ratio 1:80 (D, a red-colored line indicates the line for Δδ=0.05). The residues with Δδ greater than 0.05 are visualized as a stick model on the ribbon representation of the Dvl PDZ domain structure (E).

F    Molecular docking of Dvl binding motif (DBM) or KY-02061 to Dvl PDZ domain was analyzed by *in silico* experiments. The superimposed structure of DBM (green) and KY-02061 (yellow) on the surface of Dvl PDZ domain (gray) was visualized.

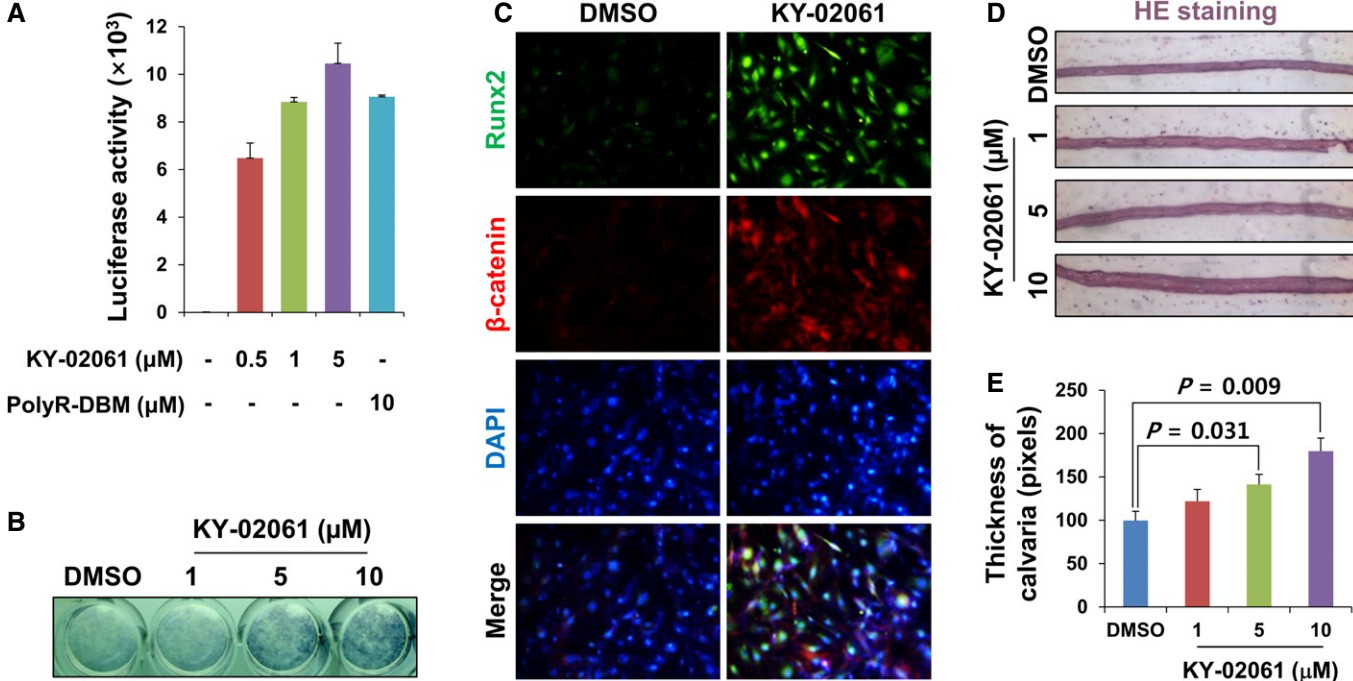

**Figure 3. Effect of KY-02061 on Wnt/β-catenin pathway, osteoblast differentiation, and *ex vivo*-cultured calvaria growth.**

A   MC3T3E1 cells were transfected with pTOPFLASH together with pCMV-β-gal. After 24 h, the cells were treated with DMSO, KY-02061 in DMSO, or 5 μM PolyR-DBM for 2 days. The luciferase activities of whole-cell lysates were measured and normalized with β-galactosidase activities. (*n* = 3)

B   Primary calvaria cells were isolated from the calvariae of 4-day-old mice (Manton *et al*, 2007) and treated with DMSO or KY-02061 in DMSO for 4 days. ALP activity levels were visualized by ALP staining.

C   Primary calvaria cells were treated with DMSO or 5 μM KY-02061 in DMSO for 14 days. The cells were subjected to immunofluorescence analyses to visualize Runx2 (green) and β-catenin (red). The cell nuclei were counterstained with DAPI (blue).

D, E   Calvariae from 4-day-old mice were cultured *ex vivo* for 7 days with KY-02061 in DMSO (D). The calvaria thicknesses were measured from the stained sections using Image Pro software (E). (*n* = 3)

Data Information: For A and E, the data are the mean ± s.d. (error bars), and significance was assessed using unpaired Student's *t*-test.

DBM-mimicking manner (Appendix Fig S3A–C and Fig 2C–E). *In silico* molecular docking analyses showed that KY-02061 potentially fitted into the groove of PDZ domain in a similar manner with DBM (Fig 2F).

KY02061 increased the activation of the Wnt/β-catenin pathway in a dose-dependent manner as revealed by the TOPflash reporter assay (Molenaar *et al*, 1996) (Fig 3A). ALP activity also increased in a dose-dependent manner in KY-02061-treated primary osteoblast cells (Fig 3B). In addition, increased levels of Runx2, an osteoblast differentiation marker (Prince *et al*, 2001), and β-catenin were observed in the cells (Fig 3C). Furthermore, the thicknesses of *ex vivo*-cultured calvaria were increased by KY-02061 treatment in a dose-dependent manner (Fig 3D and E).

**KY-02327, a metabolically stabilized KY-02061 analog, interacted with the Dvl PDZ domain similarly with KY-02061**

KY-02061 showed poor metabolic stability to be applied *in vivo* as revealed by experiments using rat liver microsomes and human hepatocytes (Appendix Table S1). To increase microsomal stability and inhibitory activity for oral administration, KY-02327 was synthesized as an analog of KY-02061 (Appendix Scheme S1). The sulfonate group was removed to increase

metabolic stability, and a 2-aminoethyl piperidyl group was attached to the carboxylic acid groups to increase the binding affinity of KY-02327 to the Dvl PDZ domain (Fig 4A). KY-02327 was more stable by 2.3-fold and 1.3-fold than KY-02061 in rat liver microsomes and in human hepatocytes, respectively (Appendix Table S1).

The $IC_{50}$ for inhibition of Dvl–CXXC5 interaction was 7.7-fold lower for KY-02327 than for KY-02061 (Figs 2B vs. 4B, and Appendix Table S1). Furthermore, KY-02327 showed enhanced effect on induction of ALP activity of osteoblast cells compared with KY-02061 (Appendix Table S1). These results suggest that KY-02327 is an improved compound in the respects of both activity and stability.

Data from fluorescence quenching experiments provide that the binding affinity between Dvl PDZ domain and KY-02327 was 8.308 ± 0.8 μM (Fig 4C). The binding of KY-02327 to Dvl PDZ domain was also confirmed by a surface plasmon resonance experiment, in which the compound binds with Dvl PDZ domain in a dose-dependent manner (Appendix Fig S4). Titration experiments revealed that one residue in βA (I250) and three residues in αB (V318, R322, and V325) of Dvl PDZ were perturbed in the interaction with KY-02327 (Fig 4D–F). *In silico* molecular docking analyses showed that KY-02327 fits into the groove of PDZ domain in a similar manner with KY-02061 (Fig 4G).

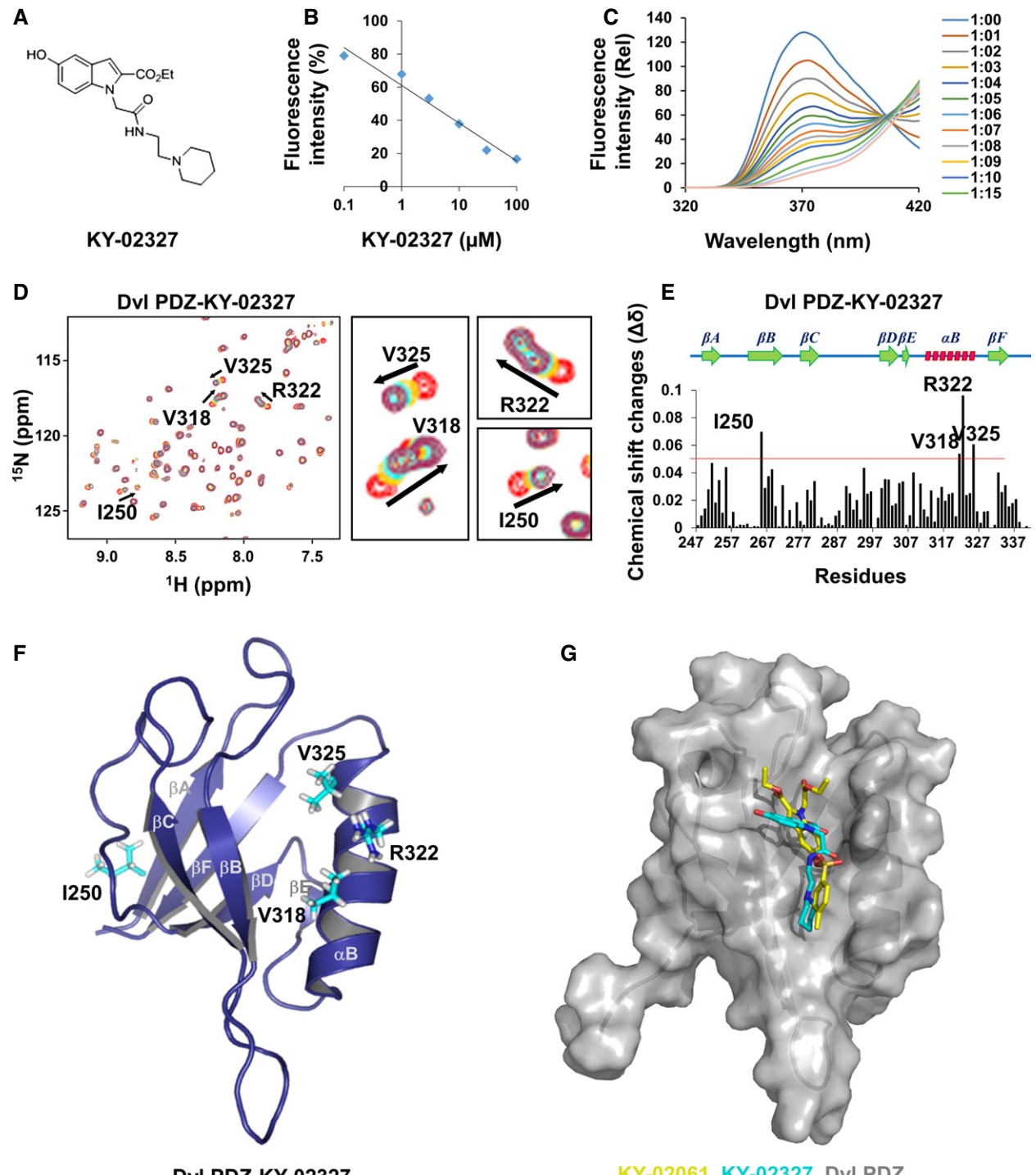

**Figure 4. Binding of KY-02327, a KY-02061 analog, on Dvl PDZ domain.**

A   The chemical structure of KY-02327.

B   A competition curve for the Dvl–CXXC5 interaction by KY-02327.

C   Fluorescence quenching plot of Dvl PDZ domain upon addition of different amounts of KY-02327. From top to bottom, final molar ratio between Dvl PDZ and KY-02327 was 1:0, 1:1, 1:2, 1:3, 1:4, 1:5, 1:6, 1:7, 1:8, 1:9, 1:10, 1:15, 1:20, and 1:25, respectively.

D–F   $^1$H-$^{15}$N-HSQC analyses were performed to analyze the interaction of $^{15}$N-labeled Dvl PDZ domain with KY-02327. The $^1$H-$^{15}$N-HSQC spectrum of different molar ratios (Dvl PDZ domain:KY-02327) is displayed as red (1:0), yellow (1:5), green (1:10), and magenta (1:20) (D, residues with meaningful chemical shift change are indicated by arrows). Plot of chemical shift changes ($\Delta\delta$) as a function of residue number in molecular ratio 1:20 (E, a red-colored line indicates the line for $\Delta\delta = 0.05$). The residues with $\Delta\delta$ greater than 0.05 are visualized as a stick model on the ribbon representation of the Dvl PDZ domain structure (F).

G   Molecular docking of KY-02061 or KY-02327 to Dvl PDZ domain was analyzed by *in silico* experiments. The superimposed structure of KY-02061 (yellow) and KY-02327 (cyan) on the surface of Dvl PDZ domain (gray) was visualized.

**KY-02327 activated the Wnt/β-catenin pathway, promoted osteoblast differentiation, and rescued BMD, bone volume, and trabecular bone structures in OVX mice**

The dose-dependent effect of KY-02327 on the transcriptional activation of β-catenin was confirmed by the TOPflash reporter system (Fig 5A). Treatment of KY-02327 on MC3T3E1 cells (a murine pre-osteoblast cell line) (Wang *et al*, 1999) increased β-catenin protein level together with Runx2 and accumulated nuclear β-catenin in a dose-dependent manner (Fig 5B and C). The mRNA levels of osteoblast differentiation markers *collagen 1a* (*Col1a*) and *osteocalcin* (*OCN*) were increased by KY-02327 treatment in a dose-dependent manner (Fig 5D and E). Overall, KY-02327 showed an inhibitory effect on the Dvl–CXXC5 interaction and activating effect on the Wnt/β-catenin pathway, resulting in promotion of osteoblast differentiation.

To evaluate the effect of KY-02327 on osteoporotic disease, KY-02327 was orally administrated to OVX mice (Fig 6A–H). Newly formed bones which labeled with calcein were decreased in the femur of vehicle-treated OVX mice compared with that of vehicle-treated Sham-operated (Sham) mice (Fig 6A, upper panels, arrows). However, the reduced bone formation of OVX mice was recovered by KY-02327 or PTH treatment (Fig 6A, lower panels, arrows). In addition, the number of osteoblasts and the calcein double-labeled surface were increased in the bones of mice treated with KY-02327 or PTH (Fig 6B and C).

The micro-computed tomography (CT) analyses showed the recovery of osteoporotic bones, indicated by the rescue of bone qualities (Fig 6D–H). The mineral contents of the bone which reduced in the femora of OVX mice were significantly increased by oral administration of KY-02327 similar to subcutaneous injection of PTH (Fig 6D). The volume of bone also increased in KY-02327- or PTH-treated OVX mice compared with vehicle-treated mice (Fig 6E). In addition, destroyed trabecular bones were rescued by KY-02327 or PTH treatment (Fig 6F). The micro-CT analyses showed the increase of trabecular bone number (Fig 6G), and the decrease of trabecular separation showing empty spaces between trabecular bones (Fig 6H) of OVX mice by KY-02327 treatment, in the similar extent of PTH treatment.

To evaluate off-target effects of KY-02327, we monitored the transcriptional regulation of target genes of major cell signaling pathways in KY-02327-treated mouse embryonic fibroblast (MEF) cells (Appendix Table S2 and Appendix Fig S5). KY-02327 significantly increased the mRNA levels of Wnt/β-catenin target genes (Appendix Fig S5A). However, the target genes of other pathways were unaffected by the treatment of KY-02327 (Appendix Fig S5B-J).

We also assessed possible toxicity of KY-02327 using conventional *in vitro* assays. KY-02327 showed no genetic toxicity in a bacterial reverse mutation assay (Maron & Ames, 1983) (Appendix Table S3). An *in vitro* cytochrome P450 inhibition assay (Marks *et al*, 2002) revealed no inhibitory effect of 10 μM KY-02327 on the liver enzymes (Appendix Table S4). KY-02327 also did not show any significant cardiotoxicity, as shown by the human ether-a-go-go-related gene (hERG) potassium channel inhibition test (Piper *et al*, 2008) (Appendix Table S5).

The weight of KY-02327-treated mice did not change significantly during the 4-week treatment (Appendix Fig S6A). No histological abnormalities were observed in the livers, lungs, or spleens of

KY-02327-treated mice (Appendix Fig S6B-D). In addition, KY-02327 treatment on normal or cancer cells derived from various tissues revealed no significant toxic effect (Appendix Fig S6E). Overall, KY-02327 enhanced new bone formation in OVX mice without noticeable toxic effects.

# Discussion

In this study, we identified and characterized small molecules stimulating bone formation through inhibition of the CXXC5–Dvl interaction, a negative feedback regulation of the Wnt/β-catenin pathway (a schematic illustration of the underlying mechanism is shown in Fig 7). The inhibitors of Dvl–CXXC5 interaction showed bone-forming effects in *ex vivo* and *in vivo* calvaria growth, and oral administration of the inhibitor recovered bone loss in post-menopausal model mice. Antibody-based drugs such as anti-Dkk-1 and anti-sclerostin antibodies (Regard *et al*, 2012) as well as the PTH-based drugs are expensive and need to be administered by injection. Therefore, development of small-molecule compounds which can be administered orally would be a valuable addition to the osteoporosis therapeutics.

NMR study revealed that both DBM and KY-02061 induced perturbation of the three residues on αB of Dvl PDZ domain, in an identical manner with sulindac (L321, R322, V325) (Lee *et al*, 2009). Sulindac induced no perturbation of residues on β-sheet of the domains (Lee *et al*, 2009); however, DBM and KY-02061 strongly interacted with β-sheet complex; both induced perturbation of a serine residue (S265) of PDZ βB domain. The improved compound, KY-02327, also induced perturbation of isoleucine residue (I250), which located on βA of the PDZ domain. These results provide evidence that inhibition of Dvl–CXXC5 interaction requires specific interaction of the inhibitors with β-sheet complex of Dvl PDZ domain, and provide an explanation for the different effects of our compounds with sulindac and its derivatives (Lee *et al*, 2009; Zhu *et al*, 2012).

The drugs stimulating the Wnt/β-catenin pathway are often suspected of inducing human cancer, because aberrant activation of the pathway is related to development and progression of various cancers (Polakis, 2000). However, in the families carrying *LRP5* gain-of-function mutations, increased incidence of cancer is not reported, and tumor developments in *Sost-* or *Dkk1*-deficient animals are not observed (Monroe *et al*, 2011). In addition, the Wnt/β-catenin pathway is inactive in osteosarcoma, and inhibition of the Wnt/β-catenin pathway contributes to tumorigenesis (Cai *et al*, 2010). Tissue specific roles of the Wnt/β-catenin pathway with cell- and organ-specific transcription factors may provide an explanation on this opposite effects (Cadigan & Waterman, 2012). In pre-osteoblasts, Tcf/β-catenin complex binds with Runx2 in their nucleus and regulates osteogenic genes specifically (Reinhold & Naski, 2007). Among the direct transcriptional target genes of Wnt/β-catenin signaling, an osteogenic gene, *Fgf18* (Reinhold & Naski, 2007), but not oncogenic genes (*c-Myc* and *cyclin D1*) (He *et al*, 1998; Shtutman *et al*, 1999), were induced by Wnt3a treatment in pre-osteoblasts (Kim *et al*, 2015). We also did not observe any sign for cancer development in *CXXC5*$^{-/-}$ mice, grown up to 1.5 years. The indirect activation of the Wnt/β-catenin signaling via release of the negative feedback loop in *CXXC5*$^{-/-}$ mice may cause less risk of cancer occurrence, compared with the direct activation of the

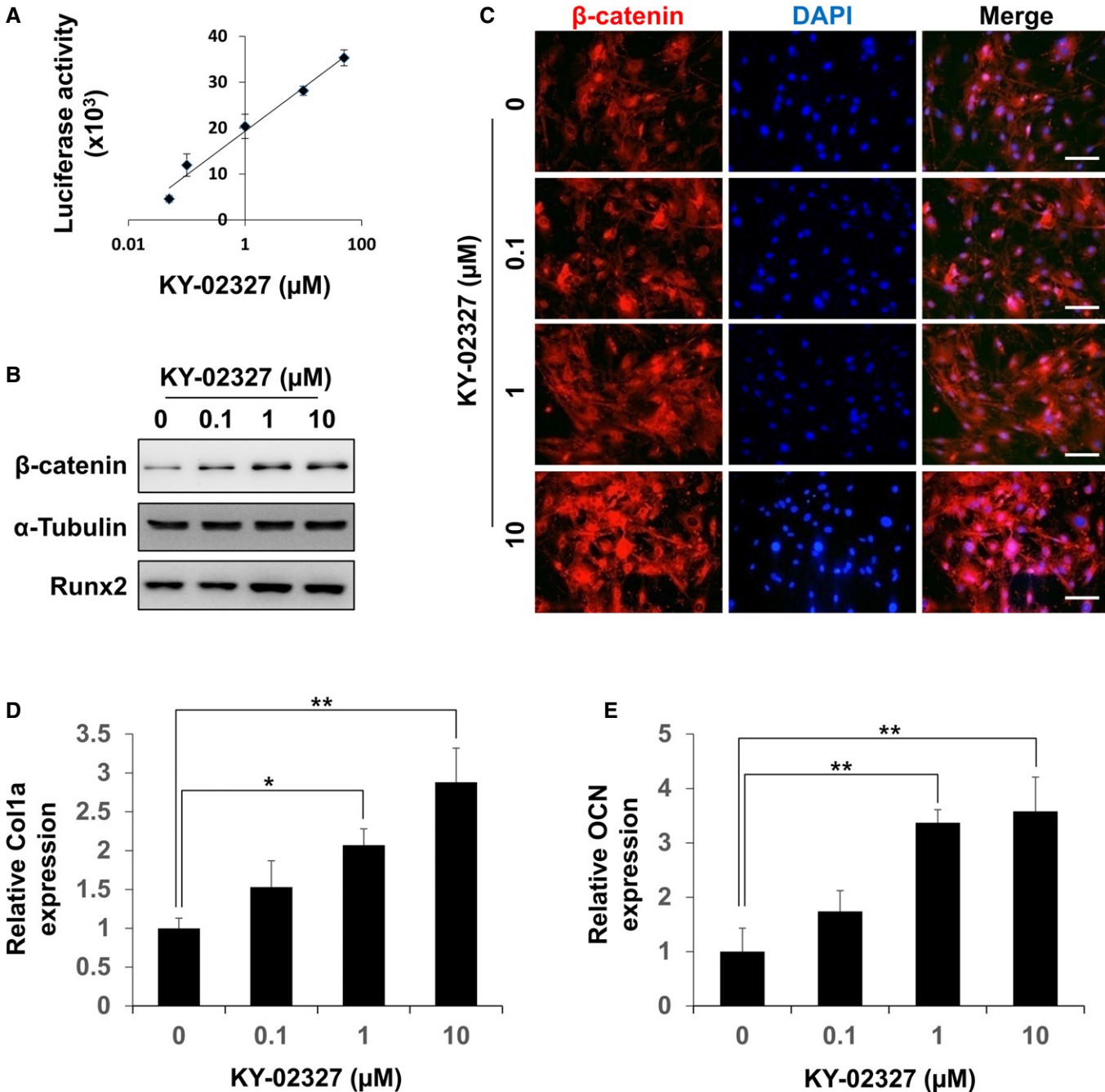

**Figure 5.  Effects of KY-02327 on Wnt/β-catenin pathway and osteoblast differentiation in osteoblasts.**

A    MC3T3E1 cells were transfected with pTOPFLASH together with pCMV-β-gal. After 24 h, the cells were treated with indicated dose of KY-02327 for 2 days. The luciferase activities of whole-cell lysates were measured and normalized with β-galactosidase activities. (*n* = 3)

B, C    MC3T3E1 cells were treated with indicated concentrations of KY-02327 for 2 days. β-Catenin and α-tubulin were detected by immunoblot (B) or immunofluorescence (C) analyses. Nuclei were counterstained with DAPI (C, blue). Scale bars, 100 μm.

D    MC3T3E1 cells were treated with indicated concentrations of KY-02327 for 14 days. The mRNA level of collagen 1a (Col1a) was measured by quantitative real-time PCR (qRT–PCR). (*n* = 3)

E    MC3T3E1 cells were treated with indicated concentrations of KY-02327 for 21 days. The mRNA level of osteocalcin (OCN) was measured by qRT–PCR. (*n* = 3)

Data Information: For A, D, and E, the data are the mean ± s.d. (error bars), and significance was assessed using unpaired Student's *t*-test.

pathway. Overall, the bone anabolic therapy activating the Wnt/β-catenin signaling via inhibition of Dvl–CXXC5 interaction may not cause any critical problem related to cancer.

In conclusion, we identified the indole-containing small molecules as candidates of orally deliverable anti-osteoporosis agents, activating the Wnt/β-catenin signaling via inhibition of Dvl–CXXC5 interaction.

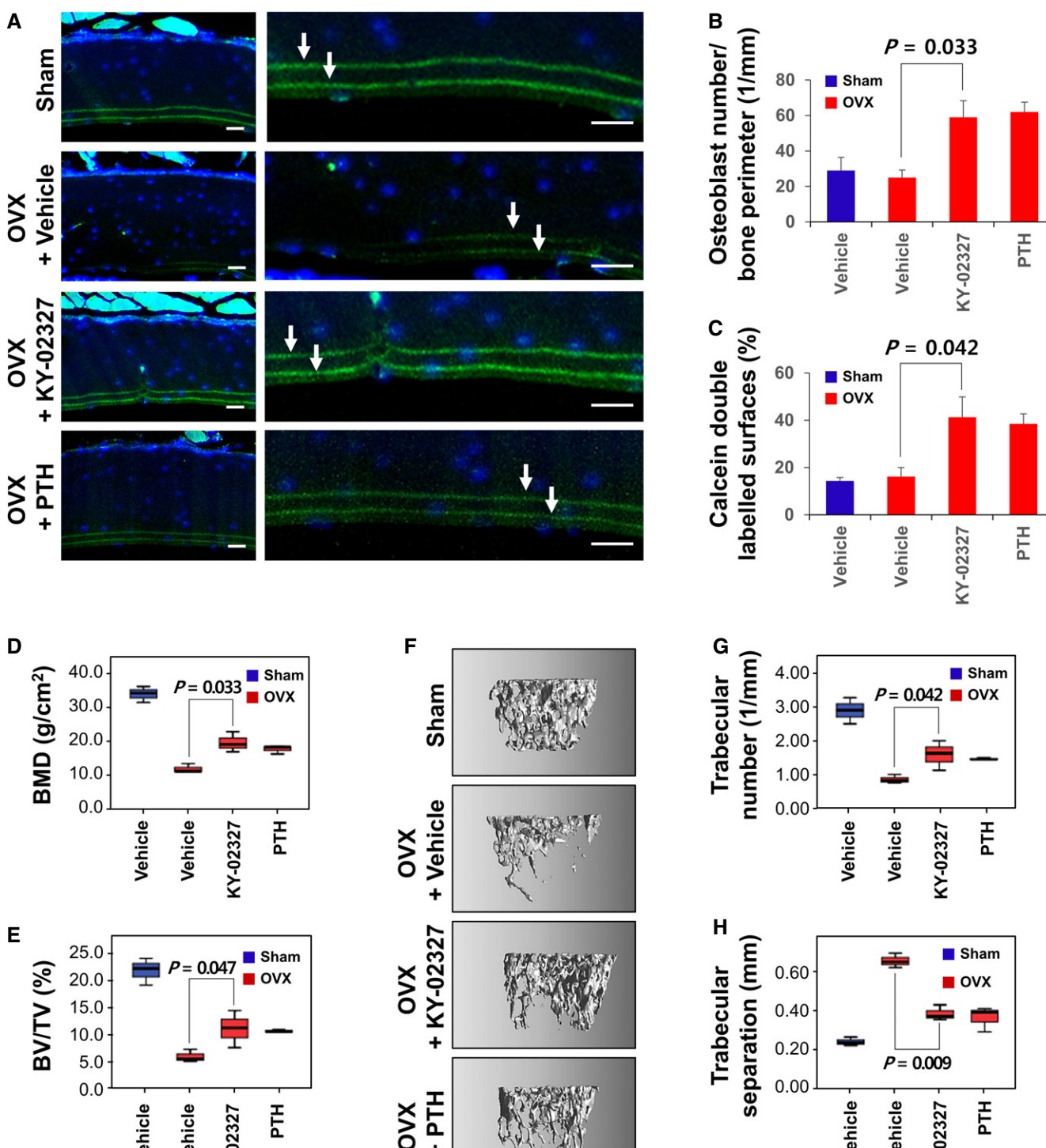

**Figure 6.   Effects of KY-02327 on OVX-induced osteoporosis model mice.**

A–H   Vehicle or 20 mg of KY-02327 per kg of animal body weight (mpk) was orally administered or 0.06 mpk of the N-terminal fragment of PTH (amino acids 1-34) was subcutaneously injected into the Sham-operated (Sham) or ovariectomized (OVX) mice on 5 sequential days per week for 4 weeks (n = 4). After calcein injection, the prepared femoral sections of the OVX mice were examined under fluorescence microscope to visualize integrated calcein (A, green, arrows). Nuclei were counterstained with DAPI (A, blue). Scale bars, 100 μm. Osteoblast numbers (B) and calcein double-labeled surfaces (C) were measured in the femoral sections. The mouse femurs were analyzed using micro-CT, and BMD (D) and BV/TV (E) were calculated from the micro-CT data. The three-dimensional images of femoral trabecular bone were reconstructed (F), and trabecular number (G) and trabecular separation (H) were calculated.

Data Information: The data are the mean ± s.d. (error bars). Significance was assessed using Kruskal–Wallis test.

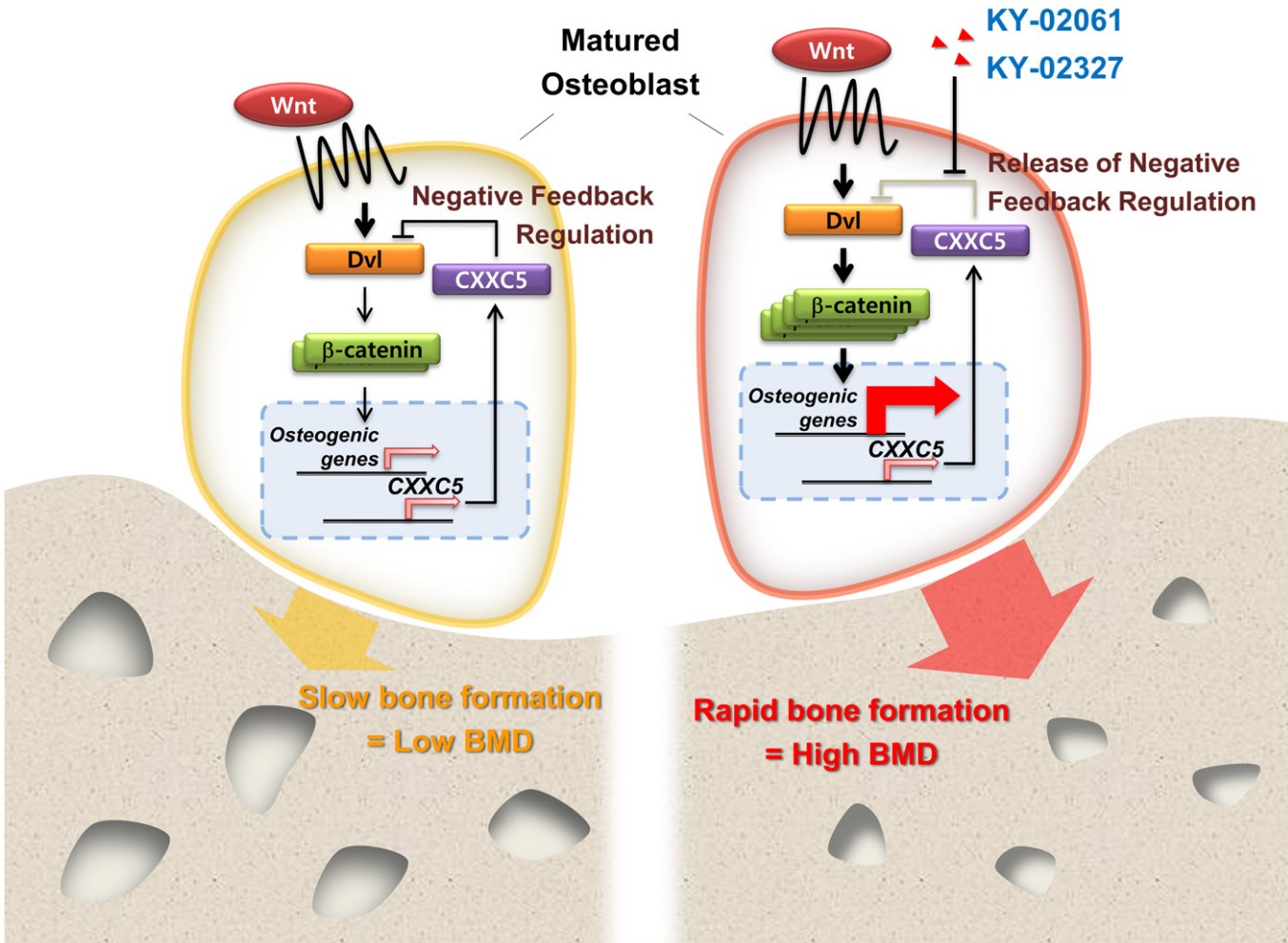

**Figure 7. A working model for stimulation of osteoblast differentiation and bone formation via small-molecule-mediated release of the negative feedback regulation (Wnt-signaling-dependent Dvl–CXXC5 interaction) of the Wnt/β-catenin signaling.**

The activation of the Wnt/β-catenin signaling at the matured osteoblasts transcriptionally induces CXXC5, and its binding to Dvl is followed by suppression of the Wnt/β-catenin signaling (left panel, suppression of the signaling marked by thin arrows) (Kim *et al*, 2015). Blockade of the Dvl–CXXC5 interaction with KY-02061 or KY-02327 releases the negative feedback regulation, resulting in high activation of the Wnt/β-catenin signaling (marked by thick arrows) and subsequent induction of osteogenic genes stimulating new bone formation and high bone mineral density.

# Materials and Methods

### Mouse calvaria extraction, *ex vivo* culture, and calvaria cell isolation

The parietal bones were isolated from the calvaria of 4-day-old mice as described previously (Reynolds *et al*, 1972). For *ex vivo* culture, each calvaria was placed on a stainless steel mesh bridge support containing α-MEM with 10% fetal bovine serum (FBS; Invitrogen, Carlsbad, CA) and 100 units/ml penicillin G streptomycin (Invitrogen) (Cha *et al*, 2014) (supplemented α-MEM). The medium was exchanged every 2 days.

Primary calvaria cells were isolated by collagenase/EDTA digestion (0.05% trypsin, 0.02% EDTA, and 0.64 mg/ml collagenase II) and plated in supplemented α-MEM (Manton *et al*, 2007). The cells were maintained at 37°C in a humidified atmosphere of 5% $CO_2$.

### Cell culture, transfections, and reporter assays

MC3T3E1 cells and primary calvaria cells were maintained in supplemented α-MEM. For the reporter assay, MC3T3E1 cells were transfected with 0.5 μg of pTOPFLASH and 50 ng of pCMV-β-gal using Lipofectamine (Invitrogen) as per the manufacturer's instruction. The relative luciferase activities are normalized with β-galactosidase activity. ALP activities were visualized using the TRACP & ALP Assay Kit (Takara, Ohtsu, Japan) according to the manufacturer's instructions. For quantitative real-time PCR (qRT–PCR), total mRNAs were extracted using TRIzol (Invitrogen), complementary DNAs were prepared using Maxime PCR PreMix (iNtRON Biotechnology, Sungnam, Korea), and the qRT–PCRs were performed using SYBR Premix Ex Taq (TaKaRa, Shiga, Japan) in the StepOnePlus Real-Time PCR machine (Thermo Fisher Scientific, Waltham, MA). The following primers were purchased from Genotech Co. (Genotech, Daejun, Korea): *COL1A*, forward: 5′-GACCGTTCTATTCCT-

CAGTGCAA-3′, reverse: 5′-CCCGGTG-ACACACAAAGACA-3′; *GAPDH*, forward: 5′-CATGGCCTTCCGTGT-TCCTA-3′, reverse: 5′-GCGGCA CGTCAGATCCA-3′; and *OCN*, forward: 5′-GTGAGCTTAACCCTG CT-TGTGA-3′, reverse: 5′-TGCGTTTGTAGGC-GGTCTTC-3′.

### Dvl–CXXC5 *in vitro* binding assay

For the Dvl–CXXC5 *in vitro* binding assay, 100 μl of 5 mg/ml purified Dvl PDZ domain was added into each well of a 96-well Maxibinding Immunoplate (SPL, Seoul, Korea) and incubated overnight in a 4°C chamber. After washing with phosphate-buffered saline (PBS), 100 μl of 10 μM PolyR-DBM (Kim *et al*, 2015) was added to each well and incubated for 4 h in a 4°C chamber. After washing with PBS 3 times, 100 μl of 2 μM small-molecule compounds in PBS was added to each well and incubated for 4 h at room temperature. After washing with PBS, the fluorescence of each well was measured using a Fluorstar Optima microplate reader (BGM Lab Technologie, Ortenberg, Germany). Small molecules used in screening were from in-house collection of Korea Research Institute of Chemical Technology (Daejeon, Korea) or were purchased from ChemDiv Inc. (San Diego, CA).

### Surface plasmon resonance

Studies on the interaction between compounds and Dvl PDZ domain were performed by surface plasmon resonance spectroscopy using the ProteOn™ XPR36 protein interaction array system (Bio-Rad Laboratories, Inc., Hercules, CA). Purified Dvl PDZ domain was immobilized on a sensor chip, and then, various concentration of KY-02327 in ProteOn PBS/Tween running buffer (PBS, pH 7.4 with 0.05% Tween 20) was flowed over the chip at a flow rate of 100 μl/min. The refractive index reflecting the mass concentration on the sensor surface is measured according to the manufacturer's instruction. Data were analyzed using ProteOn Manager Software 2.0 (Bio-Rad Laboratories, Inc.).

### Nuclear magnetic resonance spectroscopy (NMR) titration

For NMR titration, $^1$H-$^{15}$N-heteronuclear single-quantum coherence spectroscopy (HSQC) was performed using various molar ratios of $^{15}$N-labeled Dvl PDZ domain relative to the DBM peptide, KY-02061, or KY-02327. For the Dvl PDZ domain and DBM titration, the molar ratios of $^{15}$N-labeled Dvl PDZ domain to peptide ligand were 1:5, 1:10, 1:15, 1:20, and 1:25. For the Dvl PDZ domain and KY-02061 titration, the molar ratios were 1:10, 1:20, 1:40, 1:60, and 1:80. For the Dvl PDZ domain and KY-02327 titration, the molar ratios were 1:0, 1:5, 1:10, and 1:20. All spectra were processed using XWINNMR (Bruker Biospin Corp., Billerica, MA) and NMRpipe/NMRDraw software (Delaglio *et al*, 1995). Using the Sparky program, all residues in each spectrum were completely assigned, and the chemical shift perturbations were calculated using the equation $\Delta\delta_{AV} = ((\Delta\delta_{1H})^2 + (\Delta\delta_{15N}/5)^2)^{1/2}$, where $\Delta\delta_{AV}$, $\Delta\delta_{1H}$, and $\Delta\delta_{15N}$ are the average, proton, and $^{15}$N chemical shift changes, respectively. The residues showing $\Delta\delta_{AV} > 0.05$ were selected as residues participating in the interaction.

### Animal experiments and ethics statement

This animal study was approved by the Institutional Review Board of Severance Hospital, Yonsei University College of Medicine. The

### The paper explained

**Problem**

Bone anabolic agents promoting bone formation and rebuilding damaged bones would ideally overcome the limitations of anti-resorptive therapy, the current standard prescription for osteoporosis. However, the currently prescribed anabolic drugs present several adverse effects. We previously suggested the Dishevelled (Dvl)–CXXC5 interaction as a target for the development of bone anabolic agents and provided proof of principle with a Dvl–CXXC5 binding competitor peptide, which activated the Wnt/β-catenin pathway and enhanced bone-forming activities in osteoblasts and *ex vivo*-cultured calvariae. However, peptides present with several limitations as potential drugs compared to small molecules, including production costs and unfeasible oral administration.

**Results**

Here, we established a 96-well-based *in vitro* screening system, to monitor the Dvl–CXXC5 interaction in a high-throughput setting. KY-02061, a small-molecule inhibitor of the Dvl–CXXC5 interaction, was identified using the screening system and was characterized as a competitor peptide-mimicking compound enhancing bone-forming activities in osteoblasts and *ex vivo*-cultured calvariae. Structure analyses using NMR showed that KY-02061 interacted with the Dvl PDZ domain in a manner similar to the competitor peptide. We synthesized 55 KY-02061 derivatives and selected KY-02327, a metabolically stabilized compound. This compound rescued of BMD decreases and trabecular number in the ovariectomized osteoporosis mouse model when orally administered.

**Impact**

We identified and characterized orally applicable small molecules showing bone anabolic effects in a postmenopausal osteoporosis mouse model. Small molecules inhibiting the Dvl–CXXC5 interaction are potential candidates for new bone anabolic anti-osteoporotic drugs.

mice were housed in filter-topped shoebox cages with a computerized environmental control system (MJ LTD, Seoul, Korea). The care and treatment of experimental animals were performed in accordance with institutional guidelines. The room temperature was maintained at 26.5°C with a relative humidity of approximately 40–70%. The mice were given a standard maintenance diet from Dae Han Bio Link (Daejeon, Korea). For OVX mouse experiments, the surgeries were performed with 8-week-old female BL6 mice under anesthesia with avertin (2,2,2-tribromoethanol; Worthington Biochemical, Freehold, NJ) as described previously (Zahoor *et al*, 2014). After removing the hair, dorsal incisions were made into the dermal layers above both sides of the ovaries using dissection scissors. The connection between the fallopian tube and the uterine horn was cut, and the ovary was removed. The incision was then sutured with 3 single catgut stitches. Sham mice were used as controls. After 4 weeks, the OVX group was divided into three groups for oral administration of vehicle (saline containing 10% Tween-60) or 20 mg of KY-02327 per kg of animal body weight (mpk) of KY-02327, or for subcutaneous injection of 0.06 mpk of N-terminal fragment of PTH (amino acids 1–34), which were administered for 5 sequential days per week for 4 weeks. The animals were sacrificed, and the organs were harvested for analysis. For the calcein double-labeling experiment (Pall *et al*, 1987), calcein (10 mpk in 2% sodium hydrocarbonate, pH 7.4) was injected twice, on

the last day of the 2$^{nd}$ and 3$^{rd}$ week. All the mice were randomly assigned to each groups, and the investigators were blinded to the group allocation during the treatments and experiments.

### Bone mass measurements, skeletal analyses, and bone preparation

The mouse femurs were scanned using a micro-CT system (Skyscan 1072; Skyscan, Aartselaar, Belgium). The scanned image data were reconstructed to create 3D images and analyzed using the CT-Analyzing Program (Skyscan). For femoral trabecular bone analysis, the femurs were scanned with a voxel size of 10 μm, beginning at the end of the growth plate and extending proximally along the diaphysis, and 80 continuous slices (beginning at 0.1 mm from the most proximal aspect of the growth plate in which both condyles were no longer visible) were selected for analysis.

### Histochemical analysis

The femurs of OVX mice and the calvariae of 4-day-old mice were fixed in 4% paraformaldehyde in PBS. The bones were then decalcified in 10 mM EDTA and embedded in paraffin. Five-micrometer tissue sections were deparaffinized in 100% xylene followed by re-hydration with sequential 100% to 50% ethanol washes. The sections were stained with hematoxylin and eosin (H&E), and light micrographs were captured using a SMZ645 Nikon microscope (Nikon, Tokyo, Japan). The calvaria thicknesses and osteoblast numbers per bone perimeter were measured and calculated using the images using Image Pro software (Media Cybernetics Inc., Silver Spring, MD). For histomorphometric analyses, the sections were counterstained with 4′,6-diamidino-2-phenylindole (DAPI; Boehringer Mannheim, Indianapolis, IN) and mounted in Gel/Mount media (DAKO, Santa Barbara, CA). An LSM510META confocal microscope was used to capture the fluorescent micrographs of the stained sections (Carl Zeiss, Jena, Germany). Calcein double-labeled surfaces were measured from the confocal microscopic image using Image Pro software.

### Molecular docking

Molecular docking experiments was carried out using CHARMm-based DOCKER version 4.1 (Discovery Studio, Accelrys Inc., San Diego, CA; Wu *et al*, 2003). To determine binding mode of DBM and KY-02061 to the Dvl PDZ domain, the crystal structure of the Dvl PDZ domain bound with the Dapper peptide (1L6O) (Cheyette *et al*, 2002) was retrieved from the RCSB Protein Data Bank. All ligand binding poses were validated by CDOCKER score (Wu *et al*, 2003). The predicted ligand complex structures were visualized using the PyMOL program (DeLano Scientific LLC, San Carlos, CA).

### Antibodies

For Western blotting and immunostainings, the following antibodies were used: β-catenin mouse (2698, Cell Signaling Technology, Beverly, MA); Runx2 rabbit (sc-10758, Santa Cruz Biotechnology, Santa Cruz, CA); α-tubulin mouse (T6074, Sigma, St. Louis, MO); and horseradish peroxidase (HRP)-linked anti-mouse and anti-rabbit (Cell Signaling Technology). Alexa Fluor 488 and 555 donkey secondary antibodies were from Invitrogen.

### Statistics

Sample size for the *in vitro*, *ex vivo*, and *in vivo* experiments was determined according to general experimental procedures. Statistically significant differences were calculated using Student's *t*-test or Kruskal–Wallis test. For all statistical analyses, Statistical Package for the Social Sciences (SPSS 21; IBM, New Work, NY) statistical program was used.

**Expanded View** for this article is available online.

### Acknowledgements

We thank Bum Tae Kim and Sung-Youn Chang for discussion on the design and the synthesis of molecules. This work was supported by grants from the Ministry of Future Creation and Science (MFCS) of Korea through National Research Foundation (NRF); Translational Research Center for Protein Function Control (2009-0083522); and Mid-career Researcher Program (2015R1A2A1A05001873). This work is also supported by the Ministry of Knowledge Economy through Korea Research Institute of Chemical Technology (SI-0905, SI-1005, SI-1105, SI-1205, SI-1304).

### Author contributions

The project was conceptualized and supervised by KYC. The study was designed by HYK and KYC Cell and animal experiments and chemical library screening were performed by HYK and SC. NMR study was performed by JHY and WL. Synthesis of compounds and examination of their pharmacokinetic properties were performed by HJL, HL, and JNH. *In silico* molecular docking was performed by JC and KTN. The manuscript was written by HYK, JHY, HJL, JC, EJR, and KYC.

### Conflict of interest

The authors declare that they have no conflict of interest.

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
