## [Review Process File · EMBO Molecular Medicine]

Manuscript EMM-2015-05714

Small molecule inhibitors of Dishevelled-CXXC5 interaction are new drug candidates for bone anabolic osteoporosis therapy

Hyun-Yi Kim, Ms. Sehee Choi, Ji-Hye Yoon, Hwan-Jung Lim, Hyuk Lee, Jiwon Choi, Jung-Nyoung Heo, Weontae Lee, Kyong Tai No and Kang-Yell Choi

Corresponding author: Kang-Yell Choi, Yonsei University

Review timeline:	Submission date:	03 August 2015
	Editorial Decision:	04 September 2015
	Revision received:	04 December 2015
	Editorial Decision:	11 January 2016
	Revision received:	04 February 2016
	Accepted:	10 February 2016

Editor: Robert Buccione

Transaction Report:

1st Editorial Decision

04 September 2015

Thank you for the submission of your manuscript to EMBO Molecular Medicine. We have now heard back from the three Reviewers whom we asked to evaluate your manuscript.

Although all Reviewers are globally positive on your manuscript, Reviewers 2 and 3 do raise some issues that require further action. I will not dwell into much detail, but I would like to highlight the main points.

Reviewer 2 is satisfied that the main conclusions are quite well supported by the experimental data, but does feel that further histomorphometric data to demonstrate how much new osteoblast surface is generated following treatment is required to lend more conclusive support to your conclusions. This Reviewer also suggests a profiling approach to gain further insight into the potential pathways being triggered by the compounds.

Reviewer 3 suggests that KY-02327 has the potential to inhibit estrogen production thereby suppressing bone mass and would thus like you to verify blood estrogen levels after treatment. S/he also notes the discrepancy between its efficacy in vitro and in vivo and would like to attempt to clarify the reasons for this.

I would like to add that I personally concur with Reviewer 2's (and in part also Reviewer 3's) suggestion to improve English usage, clarity and readability for readers unfamiliar with both areas covered in the manuscript (bone regulation and Wnt signaling).

In conclusion, while publication of the paper cannot be considered at this stage, we would be

pleased to consider a revised version of your manuscript with the understanding that the Reviewers' concerns must be addressed with additional experimental data where appropriate and that acceptance of the manuscript will entail a second round of review.

***** Reviewer's comments *****

Referee #1 (Comments on Novelty/Model System):

The authors have developed a small molecule inhibitor of the interaction between disheveled-CXXC5. Using a combination of in vitro, cell culture and in vivo experiments they have demonstrated that they can increase osteoblast differentiation and bone formation. Importantly, they have used an in vivo bone loss model (ovariectomy) to demonstrate that the small molecule inhibitor can be given orally to increase bone formation. The inhibitor is at least as active as recombinant PTH.

It is my opinion that this is an important advance in the treatment of bone loss/osteoporosis. There is currently no oral therapy available increase of bone formation. Currently available oral drugs inhibit bone resorption and do not increase bone formation.

Referee #1 (Remarks):

This is a well done study that demonstrates the development and utility of an orally administered agent that increases bone formation. The reviewer has no objections to any of the methods used, the statistical analysis thereof and the conclusions drawn by the authors.

Referee #2 (Comments on Novelty/Model System):

The authors have used a variety of techniques to examine the ability of small molecules to disrupt the interaction of disheveled protein and a disheveled binding protein (CXXC5) on the Wnt/catenin signaling pathway and ultimately bone formation in several in vitro and in vivo models. The fact that the authors have found small molecules that appear to work through the release of the negative regulation Wnt/catenin signaling is impressive although the structure of the lead molecule might render it unstable.

While the authors have demonstrated an increase in bone volume, I would want to also see bone histomorphometric data on the amount of active osteoblast surface in the bones of animals treated with the compounds - this would provide further evidence to support the comments made in the manuscript concerning the linkage between activation of the Wnt/catenin pathway, osteoblast recruitment and osteoblast activity (i.e., bone formation). The narrative in the introduction and additional sections needs some attention to help the reader (expert or non-specialist) understand the subject matter and the context for the study. The use of an array of techniques - including a molecular structure function examination - may make the paper eligible for publication in EMBO Mol Med.

Referee #2 (Remarks):

Interesting paper that uses a variety of in vitro and in vivo models to assess the ability of small molecules to disrupt a protein/protein interaction that negatively regulates the Wnt/catenin pathway (and bone formation). The experiments appear to have been appropriately designed and well controlled. The authors describe a series of results - including a molecular structure/function analysis of the mechanism of disruption by the aforementioned small molecules. The in vivo model data is compelling and I would want to the authors to use bone histomorphometry to show how much new osteoblast surface is created following treatment and how much of that surface would be considered as an active bone formation surface - this would add further support to the comments made in the introduction regarding the impact of the Wnt/catenin signaling on osteoblast activity and bone formation. Additionally, have the authors considered submitting the compounds to a profiling service such as BioSeek or Genometry's L1000 assay to gain further information about

target and off-target engagement of these molecules? That profiling data could add further weight to the specificity of the pathway involvement and potentially give some insight into the toxicity induced by some of the compounds - this would also give another set of important reference information for the primary data.

The paper needs a re-write of certain passages (introduction as an example) to improve the grammar and establish a greater clarity of background context and to define the aims of the study for readers who are familiar with the concepts of bone regulation and turnover but who may not be aware of the nomenclature and terms that describe the individual regulators of Wnt/catenin signaling (i.e., CXXC5 is a disheveled binding protein).

Referee #3 (Comments on Novelty/Model System):

In the ovariectomized mice model the compound KY-02327 has very modest effect in restoring bone loss. Yet the same compound has more profound effect in vitro (Fig 3). This may be because canonical WNT Signaling has been shown to inhibit FSH (follicle stimulating hormone) mediated steroidogenesis in rodent granulosa cells. This means that compound KY-02327 has potential to inhibit estrogen production and suppress bone mass. Therefore, it would seem logical to estimate blood estrogen concentration in the injected animals and controls.

In vitro cell studies could be improved by looking at the expression of a number of key downstream osteogenic transcripts, such as osterix, collagen and osteocalcin.

Referee #3 (Remarks):

The work presented is an extension of previously published concept but it has obvious translational implications. I highlight some of the main points about the manuscript:

1. Canonical WNT Signaling has been shown to inhibit FSH (follicle stimulating hormone) mediated steroidogenesis in rodent granulosa cells. This means that compound KY-02327 has potential to inhibit estrogen production and suppress bone mass. Therefore, it would seem logical to estimate blood estrogen concentration in the injected animals and controls.
2. In addition to effect on alkaline phosphatase it would be informative to see effect of KY-02327 on MC3T3E1 cells proliferation and expression of key transcripts, such as collagen 1a and osteocalcin.
3. In the ovariectomized mice KY-02327 has very modest effect in restoring bone loss. Yet the same compound has more profound effect in vitro (Fig 3). Could this be explained by in vivo inhibition of steroidogenesis as explained in 1 above?
4. Manuscript length could be reduced by at least one third by avoiding repetition and omitting methods given in the results.
5. The number of replicates of in vitro cell experiments should be given.

1st Revision - authors' response

04 December 2015

RESPONSE TO REVIEWERS

REVIEWER #1

We greatly appreciate the reviewer's interest and comments on our work.

REVIEWER #2

1. The reviewer requested bone histomorphometry analysis for the *in vivo* experiments.

As the reviewer requested, we performed bone histomorphometric analyses on KY-02327- or PTH-treated ovariectomized (OVX) mice. We quantified the number of osteoblasts and the calcein double-labelled surface on the bone. Both the number of osteoblasts and the calcein double-labelled surface were increased approximately 2-fold by KY-02327 treatment. This result was similar to that observed with PTH treatment (Alexander et al., J Bone Miner Res. 2001, 16:1665-73). We provided these new data in Fig. 6, B and C and described the results (page 13, line 16-18) in the revised manuscript.

2. The reviewer suggested to evaluate possible off-target effects and toxicity of KY-02327.

KY-02327 exhibited a specific inhibitory effect on the Dvl-CXXC5 interaction *in vitro*, and the KY-02327-Dvl PDZ complex structure was determined, suggesting that KY-02327 may specifically inhibit protein-protein interaction. However, to address the reviewer's concern of off-target effects further, we monitored the transcriptional regulation of target genes of major cell signaling pathways in mouse embryonic fibroblast cells after 12 hours of treatment with KY-02327 (Table S2, Fig. S5 and page 14, line 11-17). KY-02327 significantly increased the mRNA levels of Wnt/b-catenin target genes (Fig. S5A). However, the target genes of other pathways were unaffected by the treatment of KY-02327 (Fig. S5B-J). These results show the specificity of KY-02327 on the Wnt/b-catenin pathway.

We also assessed the possible toxicity of KY-02327 using conventional *in vitro* tests, including the Ames test, a CYP inhibition assay, and hERG binding assays. KY-02327 did not show significant toxicity in these genetic (Table S3), hepatic (Table S4), and cardiac (Table S5) tests. We have provided these data (Table S3-S4) and described the results in the revised manuscript (page 14, line 18- page 15, line 6).

In addition, we did not observe any significant changes in the weight of OVX mice treated with KY-02327 (Fig. S6A). Organs from drug-treated animals did not show significant histological abnormalities (Fig. S6, B-D and page 15, line 7-10).

Finally, treatment with KY-02327 did not significantly alter the proliferation of various types of normal or tumor cells (Fig. S6E and page 15, line 10-13). Overall, KY-02327 did not cause significant toxicity *in vivo* or *in vitro*. However, further chemical synthesis of analogs to develop a compound optimized for pharmacological characteristics and functionality will be required in the process of drug development.

3. The reviewer requested to improve English usage, clarity, and readability for readers unfamiliar with bone mass regulation and Wnt signaling.

We have added an extra explanation of bone mass regulation (page 3, line 5-10) and Wnt signaling (page 4, line 16–page 5, line 4). Further, the English usage of the revised manuscript is improved.

REVIEWER #3

1 & 3. The reviewer raised a concern with inhibition of estrogen production by KY-02327 treatment. He or she suggested that the inhibition could be explained by the difference between *in vivo* and *in vitro* effects of KY-02327.

Recent studies have shown that both canonical and non-canonical Wnt pathways regulate granulosa cells and steroidogenesis in the ovary (Abedini *et al.*, 2015; Stapp *et al.*, 2014). In our OVX mouse experiments, however, we completely removed both ovaries, meaning that no granulosa cell existed and no estrogen production occurred in these animals. Therefore, Wnt-mediated regulation of estrogen production would have no effect on our animal experiments. However, we will carefully consider this point in the development of KY-02327 and its derivatives as a human therapeutic agent.

2. The reviewer requested to show the effects of KY-02327 on proliferation and differentiation of MC3T3E1 cells.

The effect of KY-02327 on proliferation of MC3T3E1 was monitored by MTT assay. The results showed that KY-02327 did not significantly affect the proliferation of MC3T3E1 cells (Fig. S6E). Additionally, the proliferation of murine primary osteoblasts (OB) was unaffected by KY-02327 treatment (Fig. S6E). Meanwhile, quantitative real-time PCR analyses on KY-02327-treated MC3T3E1 cells showed that mRNA levels of *collagen 1a (Coll1a)* and *osteocalcin (OCN)* increased

after KY-02327 treatment in a dose-dependent manner (Fig. 5, D and E). These results are described in the revised manuscript (page 13, line 6-10).

4. The reviewer suggested to reduce the manuscript length by avoiding repetition and omitting methods given in Results.

As the reviewer requested, we have edited our manuscript to make it more concise.

5. The reviewer pointed that the numbers of replicates of cell experiments should be given.

As the reviewer requested, we have provided the replicate numbers for all cell experiments in the figure legends of the revised manuscript.

References

Alexander JM, Bab I, Fish S, Müller R, Uchiyama T, Gronowicz G, Nahounou M, Zhao Q, White DW, Chorev M, Gazit D, Rosenblatt M. Human parathyroid hormone 1-34 reverses bone loss in ovariectomized mice. *J Bone Miner Res.* 2001, 16:1665-73.

Abedini A, Zamberlam G, Boerboom D, Price CA. Non-canonical WNT5A is a potential regulator of granulosa cell function in cattle. *Mol Cell Endocrinol.* 2015, 403:39-45.

Stapp AD, Gómez BI, Gifford CA, Hallford DM, Hernandez Gifford JA. Canonical WNT signaling inhibits follicle stimulating hormone mediated steroidogenesis in primary cultures of rat granulosa cells. *PLoS One.* 2014, 9:e86432.

2nd Editorial Decision

11 January 2016

Dear Prof. Choi,

Thank you for the submission of your manuscript to EMBO Molecular Medicine and please accept our apologies for the unusual delay, due also to the concomitant holiday season.

We have now heard back from the two Reviewers whom we asked to evaluate your manuscript.

As you will see the Reviewers are now satisfied with your manuscript and I am thus prepared to accept your manuscript for publication pending my following editorial requests:

1) Please note the comment by Reviewer 2. I agree with him/her that the manuscript needs some work to improve delivery, comprehension and impact of your work. Your manuscript features much valuable data including structural biology and should read as a comprehensive analysis of molecular structure function together with the molecules, but it is indeed lacking cohesion and fluency. Some transition statements in the narrative to link the various approaches in a more cohesive manner would enormously help.

2) As per our Author Guidelines, the description of all reported data that includes statistical testing must state the name of the statistical test used to generate error bars and P values, the number (n) of independent experiments underlying each data point (not replicate measures of one sample), and the actual P value for each test (not merely 'significant' or 'P < 0.05').

***** Reviewer's comments *****

Referee #2 (Remarks):

Please seek out the help of a medical writer who can review and join together more effectively the narrative in the paper. Science is strong and experiments have been well executed but it needs some work to make it flow a lot easier for the reader.

Referee #3 (Comments on Novelty/Model System):

The authors have addressed the points I raised and the manuscript is now in an acceptable form.

Corresponding Author Name: Kang-Yell Choi

Manuscript Number: EMM-2015-05714